# In Utero Exposure to Hormonal Contraception and Mortality in Offspring with and without Cancer: A Nationwide Cohort Study

**DOI:** 10.3390/cancers15123163

**Published:** 2023-06-13

**Authors:** Lina Steinrud Mørch, Mads Gamborg, Caroline Hallas Hemmingsen, Charlotte Wessel Skovlund, Susanne Krüger Kjær, Marie Hargreave

**Affiliations:** 1Cancer Surveillance and Pharmacoepidemiology, Danish Cancer Society Research Center, Strandboulevarden 49, 2100 Copenhagen, Denmark; maga@cancer.dk (M.G.); cws@cancer.dk (C.W.S.); 2Virus, Lifestyle, and Genes, Danish Cancer Society Research Center, Strandboulevarden 49, 2100 Copenhagen, Denmark; cahe@cancer.dk (C.H.H.); susanne@cancer.dk (S.K.K.); mariehar@cancer.dk (M.H.)

**Keywords:** pharmacoepidemiology, in utero exposure, hormonal contraception, child mortality, childhood cancer, prognosis

## Abstract

**Simple Summary:**

Hormonal contraception is widely used among reproductive-aged women. In spite of the high effectiveness of hormonal contraception, some users become pregnant, probably due to irregular use. We previously reported an increased morbidity of childhood cancer in a nationwide cohort of offspring exposed in utero to maternal use of hormonal contraception. However, it remains unknown if mortality is increased in offspring with and without cancer after in utero exposure to hormonal contraception. The present study indicates that in utero exposure to hormonal contraception has an influence on long-term child mortality and survival after a diagnosis of leukemia. These novel findings have potential use in guidelines for hormonal contraception use in relation to pregnancy and expand our understanding of the etiology and prognosis of childhood leukemia.

**Abstract:**

Approximately 400 million women of reproductive age use hormonal contraceptives worldwide. Eventually, pregnancy sometimes occurs due to irregular use. Use in early pregnancy is found to be associated with child morbidities including cancer, the main reason for disease-related death in children. Here, we add the missing piece about in utero exposure to hormonal contraception and mortality in offspring, including assessments of prognosis in children with cancer. In utero exposure to hormonal contraception may be associated with death since we found a hazard ratio (HR) of 1.22 (95% confidence interval (CI) 1.01–1.48) compared to children of mothers with previous use. The HRs were 1.22 (95% CI 0.99–1.13) for oral combined products and 2.92 (95% CI 1.21–7.04) for non-oral progestin-only products. A poorer prognosis was also found in exposed children with leukemia (3.62 (95% CI: 1.33–9.87)). If causal, hormonal contraception in pregnancy seems detrimental for offspring health and a marker of poorer prognosis in children with leukemia.

## 1. Introduction

Hormonal contraception use has increased substantially in recent decades, and an estimated 407 million women of reproductive age (15–49 years) use hormonal contraceptives worldwide (2019) [1]. In Denmark, the vast majority of women of reproductive age use some kind of hormonal contraception (i.e., 50% of women between 15 and 19 years, 60% between 20 and 24 years, 40% in women aged 25–29 years, 31–34% until 45 years, and 23% of women aged 45–49 years of age) [2]. Although hormonal contraceptives are highly effective at preventing pregnancy, unintended pregnancy does occur, primarily due to suboptimal adherence and use [3]. In utero exposure to hormonal contraception is rare; however, given the prevalent use of hormonal contraception numerically, many children will eventually be exposed. Thus, it is important to know the overall health consequences for children born after in utero exposure to hormonal contraception.

It is known that in utero drug and hormonal exposures (e.g., thalidomide and diethylstilbestrol) can cause not only increased morbidity but also increased mortality in the offspring [4,5,6,7,8]. Diethylstilbestrol, a nonsteroidal estrogen medication, was prescribed to pregnant women in the early 1940’s to prevent miscarriages, and approximately 10 million women were exposed to this drug between 1940 and 1970 [6]. The use of diethylstilbestrol declined after several studies observed long-term health consequences, including several types of cancer [7,9,10,11,12,13], but also other adverse health outcomes [14,15] among the offspring of women prescribed diethylstilbestrol during pregnancy. We previously reported that children exposed in utero to hormonal contraception were at an increased risk of leukemia [16]. Globally, childhood cancer is the leading cause of disease-related death in children [17,18]. However, there is a lack of assessments of child mortality among children exposed in utero to hormonal contraception. Moreover, it could be speculated that leukemias with a hormonal etiology may have a different prognosis than leukemias that develop unrelated to in utero exposure to hormonal contraception [16]. The incidence of childhood cancer in Denmark has increased, especially since 1977 [19]. Overall, the two most common cancer types in children are leukemia (26.5%) and central nervous system (CNS) tumors (23.4%) [19,20]. The Nordic countries have the highest survival of childhood cancers across Europe [21]. However, the five-year survival for all malignant childhood cancer in Denmark during the period 1990–1994 was 71.6% [21], which is lower than the other Nordic countries (≥75%).

To our knowledge, no studies have assessed in utero exposure to hormonal contraception and mortality in offspring with and without cancer. In the present study, we aimed to assess overall child mortality as well as prognosis in children with cancer after in utero exposure to hormonal contraception.

## 2. Materials and Methods

### 2.1. Study Design

This study is a nationwide population-based, retrospective cohort study based on registry data.

### 2.2. Data Sources

This study used prospectively collected data by use of the unique personal identification number (PIN) given to all citizens in Denmark since 1968, and we linked newborns and their respective parents [22]. From population-based nationwide registries, we achieved information on vital status, date of migration and death as well as baseline characteristics of the children and their parents, i.e., from the Danish Civil Registration System [23], the Danish National Patient Registry [24], the Danish Medical Birth Registry [25] and Statistics Denmark [26]. To assess death in the children who developed cancer, the Danish Cancer Registry [27] was additionally used to identify populations of children who developed childhood cancer [28]. Exposure information about maternal hormonal contraception use was from the Danish National Prescription Registry [24]. Information on the characteristics of the children and their parents was from the above-mentioned registers and Statistics Denmark.

### 2.3. Validity and Completeness

The Danish Civil Registration System provided easily accessible, complete and linkable information of each person and their children [29]. We had virtually no loss to follow-up for persons with residence in Denmark [23]. The Danish Medical Birth Registry provided data of high validity and completeness with information about gestational length, which is accurate within one week for 87% of records [30]. This enables linkage with the prescription register on the timing of pregnancy in relation to exposure. The Danish National Prescription Registry contains data on all prescriptions dispensed for Danish citizens with linkage through the unique PIN number. Use of the reimbursement-driven record keeping and automated barcode-based data entry provides high quality data, including detailed information on the prescribed drugs [24]. The Danish Cancer Registry is estimated to be 95% to 98% complete [31]. Death is recorded in the Danish Civil Registration System, which provides complete information on death [29]. Information was almost complete on potential confounders (missing information reported in Table 1).

### 2.4. Study Populations

We followed all newborns in Denmark between 1 January 1996 and 31 December 2018 from birth to 31 December 2018. Children with missing or implausible gestational length were excluded. To ensure available information on maternal hormonal contraception use for a minimum of one year before child birth, children born before 1996 were not included (the Danish National Prescription Register includes information from 1995 onwards) [24]. Finally, children born by mothers with missing information on maternal age were excluded (Figure 1).

We identified three subpopulations: (1) children who developed any type of childhood cancer, (2) children who developed leukemia (3) and children who developed CNS tumors (Figure 1).

### 2.5. Measures and Variables

#### 2.5.1. Exposure

Exposure to hormonal contraception use was defined according to the timing of maternal hormonal contraception use in relation to pregnancy. The timing of use was identified using the dates of redeemed prescriptions on hormonal contraceptives from the Danish National Prescription Registry [24] and the date of onset of pregnancy. The latter was defined by subtracting the gestational length from the date of birth. Information on gestational length was extracted from the Danish Medical Birth Registry [25]. The primary exposures of interest were maternal hormonal contraception use (1) “during pregnancy” (i.e., in utero exposure) or (2) “recent use” (i.e., use up to pregnancy start). Exposure in utero was defined as maternal redeemed prescriptions of hormonal contraception during the first or second trimester of pregnancy. “Recent use” of combined contraceptives, oral progestin-only products and emergency contraceptives was defined as prescriptions redeemed 3 months or less before the start of pregnancy. “Recent use” of non-oral progestin-only products was defined as redeemed prescriptions as follows: (1) Hormonal intrauterine device (IUD), ≤3 (Jaydess), ≤5 (Mirena, Kyleena) or ≤6 (Levosert) years before pregnancy start; (2) hormonal implant (Implanon, Nexplanon), ≤3 years before pregnancy start; and (3) injectable hormonal contraceptives (Depo-Provera), ≤1 year before pregnancy start. “Previous use” was defined as redeemed prescriptions before recent use. To mitigate confounding by unknown maternal characteristics, we used children exposed to maternal previous use as the reference group [32]. This group was used as a reference group throughout all analyses. Maternal use of hormonal contraception was analyzed according to any type and specific types, i.e., combined hormonal contraception, including estrogen and progestin, and progestin-only products. In addition, the hormonal contraceptive products were further subdivided by route of administration into oral and non-oral types. Anatomical Therapeutic Chemical (ATC) codes and the classification of hormonal contraception are presented in Appendix A.

#### 2.5.2. Other Variables

Other variables included information on the children and their parents, i.e., year of birth, sex, origin, birth order, parental age at birth and education, maternal infertility, maternal body mass index, maternal smoking and perinatal factors (Appendix A).

#### 2.5.3. Outcome

The primary outcome of interest was death assessed among all children and among children who developed any type of cancer, leukemia or a CNS tumor. Information on the death of the children was obtained from the Danish Civil Registration System [23].

#### 2.5.4. Other Measurements

Childhood cancer was defined as cancer diagnosed before the age of 20 years according to the International Classification of Childhood Cancer, third edition (ICCC-3) [28].

### 2.6. Statistical Analysis

Hazard ratios (HRs) and 95% confidence intervals (CIs) were calculated using the Cox proportional hazards model for death among children exposed to hormonal contraception (1) in utero and (2) recently up to pregnancy compared with children of mothers who used hormonal contraception previously. Follow-up was from the date of birth to date of death, date of emigration or end of follow-up 31 December 2018, whichever came first. The statistical models included the exposure variable of interest and the a priori selected potential confounders (maternal age (<28, 28–31, 32–35, >35) and calendar year of birth (1996–1999, 2000–2004, 2005–2009, 2010–2014, 2015–2018)) as strata in the models to fulfill the proportional hazards assumption.

In the case of fatality analyses, the children with the cancer in question (i.e., of any cancer, leukemia or a CNS tumor) were followed from the date of cancer diagnosis to date of death, date of emigration or end of follow-up 31 December 2018, whichever came first. The Cox proportional hazards models included the exposure variable of interest and the a priori selected potential confounders maternal age and calendar year at birth (in the above-mentioned categories) as well as the age at diagnosis (<1 year old, 1–5 years old, 6–10 years old and >10 years old) included as strata.

Other potential confounders, i.e., parental education level, maternal infertility diagnosis, birth order, parental age at birth and origin, were assessed by change in estimate methods (complete case analyses) categorized as shown in Table 1. None of the above-mentioned potential confounders changed the risk estimate for death by >10%. Perinatal factors were considered mediators and were not included in the final models. For all analyses, a significance level of 0.05 (5%) was applied with two-tailed *p*-values. The analyses accounted for correlation between siblings by use of a robust variance estimate adjusting for within-cluster correlation [33]. Tests for the proportional hazards assumption were fulfilled for all models and were based on Schoenfeld residuals. Mortality rates (MRs) with 95% CIs were calculated with Poisson regression adjusted for the year of birth and maternal age at birth. We conducted a sensitivity analysis to explore the extent to which unmeasured confounding may have affected our findings based on the E-value [34]. Analyses were performed with the statistical software Stata SE, version 14.2 (StataCorp).

The project is registered with the Danish Cancer Society in agreement with the General Data Protection Regulation, and authorization from the appropriate authorities was obtained for the secondary research of the registry health data reported in this study. According to Danish legislation, it is not necessary to seek ethics approval or informed consent for studies based on registry data.

## 3. Results

We followed 1,426,392 live-born children for a median of 11.3 years (interquartile range (IQR) 5.4–17.1, total person-years: 16,099,288) (Figure 1). During this time, 7390 children died.

A total of 3027 children developed childhood cancer, and of these, 823 children developed leukemia and 757 a CNS tumor (Figure 1). The population of children who developed any type of cancer was followed for a median of 5.1 years (IQR 1.9–10.2, total person-years: 19,658.8), children with leukemia for 6.8 years (IQR 2.4–12.5, total person-years: 6306.9) and children with CNS tumors for 4.2 years (IQR 1.5–8.6, total person-years: 4262). During this time, 388 children died (94 after leukemia and 141 after a CNS tumor diagnosis).

Of the 1,426,392 children followed, 17,513 (1.2%) were children exposed in utero to hormonal contraception, 154,381 (10.8%) were children of mothers who had used hormonal contraception recently before pregnancy, and 952,800 (66.8%) were born to mothers who had used hormonal contraception previously. Finally, 301,698 (21.2%) were children of mothers who had not used hormonal contraceptives. Of the 3027 children who developed cancer, 43 (1.4%) were children exposed in utero to hormonal contraception, 304 (10%) were children of mothers with a recent use, 1864 (61.6%) were children of mothers with a previous use, and 816 (27%) were children of mothers who had not used hormonal contraception.

The characteristics of all the children and their parents are shown in Table 1. Furthermore, the characteristics of the children who developed any cancer, leukemia or a CNS tumor are shown in Appendix A. The perinatal characteristics are shown in Appendix A.

### 3.1. Long-Term Child Mortality

Children exposed in utero to any type of hormonal contraception had a higher HR for death (1.22, 95% CI 1.01–1.48; *p* = 0.040), whereas children born by mothers who used hormonal contraception recently up to pregnancy had a similar HR of death (1.02, 95% CI 0.94–1.11; *p* = 0.613) compared to children of mothers who had used hormonal contraception previously (Table 2).

Children exposed in utero to oral combined contraceptives had a HR of death of 1.22 (95% CI 0.99–1.50; *p* = 0.064), while the HR was 2.92 (95% CI 1.21–7.04; *p* = 0.017) in children exposed in utero to non-oral progestin-only products. The HR of death in children exposed in utero to emergency contraceptives was 1.31 (95% CI 0.55–3.27, *p* = 0.491). Non-oral combined and oral progestin-only products were not associated with death in the children (Table 3). Of note, the estimation of some HRs was not possible due to few observations.

### 3.2. Case Fatality

Among children who developed any type of cancer during follow-up, the HRs of death were 1.49 (95% CI 0.70–3.17) among children exposed in utero to hormonal contraception and 1.18 (95% CI 0.85–1.64) in children exposed to maternal recent use compared to children of previous users. Among children with leukemia, the HRs were higher: 3.62 (95% CI 1.33–9.87, *p* = 0.012) for in utero exposure and 1.35 (95% CI 0.69–2.63, *p* = 0.379) for exposure to recent use. Generally, children who developed CNS tumors were not at a higher risk of death after exposure to maternal use of hormonal contraception in utero or recently up to pregnancy (Table 4).

### 3.3. Types of Hormonal Contraception

In utero exposure to oral products (combined or progestin-only) showed similar HRs of death in children with any childhood cancer (Appendix A). Among children with leukemia, the HRs were, however, higher in children exposed in utero to oral progestin-only products than in children exposed in utero to oral combined contraception (8.00, 95% CI 1.44–44.39 versus 2.78, 95% CI 0.97–7.97) when compared with maternal previous use. In children with CNS tumors, the HR of death was 0.76 (95% CI 0.10–5.67) after in utero exposure to oral combined products.

### 3.4. Sensitivity Analyses

For exposure in utero to hormonal contraception, the observed HR for mortality in the offspring (1.22) could be explained by an unmeasured confounder associated with both the exposure and the outcome by a HR of at least 1.74. For in utero exposure to non-oral progestin-only products, the observed HR for child mortality (2.92) needed an unknown confounder to be associated with both the exposure and the outcome by a HR of at least 5.29 to explain the association.

Among children with cancer, the HR for death (1.49) in the exposed in utero to any type of hormonal contraception needed an unknown cofounder associated with the exposure and outcome by a HR of at least 2.34 to explain the observed association. Among children with leukemia, the HR for death (3.62) in the exposed in utero to any type of hormonal contraception would need an unknown confounder associated with both the exposure and the outcome with a HR of at least 3.08 to explain the observed association.

## 4. Discussion

The overall child mortality was increased in children exposed to any type of hormonal contraception in utero, while children exposed to recent maternal use showed similar mortality as children of mothers with a previous use. In addition, in children with leukemia, in utero exposure to hormonal contraception was associated with a higher risk of death than in children of mothers with a previous use. Exposure to non-oral progestin-only products showed a stronger association than exposure to oral combined products. However, although statistically significant, the observed frequencies are small; therefore, caution is needed in interpreting the results.

This study is the first to assess overall long-term child mortality and prognosis following cancer in children exposed to hormonal contraception in utero. Even though in utero exposure to hormonal contraception is rare, millions of exposed pregnancies will occur worldwide due to the high number of women using hormonal contraception [1], the effectiveness not being 100% and irregular use [35]. The current finding indicates that exposure in utero to hormonal contraception could have an overall detrimental influence on long-term child health. However, exposure to maternal “recent” use of hormonal contraception was reassuringly not associated with increased long-term child mortality compared to previous use.

Oral combined contraception includes both estrogen and progestin and is the predominant type of hormonal contraception used [2]. Encouragingly, the strongest associations were observed in children exposed to products that are less commonly used, i.e., the non-oral progestin-only products. The difference in the findings between oral and non-oral progestin-only products can be explained by the differences in the type of progestin and route of administration (Appendix A).

Very few studies on hormonal contraception and fetal toxicity in humans exist. Animal studies on fetal exposure to estrogen and progesterone have, however, shown detrimental effects on health and disease. A study of male mouse fetuses reported that in utero estrogen exposure caused serious deformities [36], and in utero progesterone exposure has been reported to alter fetal pituitary and testicular function and steroid profile in ovine males [37]. However, concerning human studies, an older study [38] reported that Depo-Provera (injectable hormonal contraceptive) is associated with an increased risk of congenital malformation in children exposed in utero to the drug, while two studies from Thailand found higher rates of neonatal and infant mortality [39] and an increased risk of chromosomal anomalies and major malformations [40] in children exposed in utero to Depo-Provera compared with controls not exposed in utero to the drug. In humans, sex hormones are considered potent carcinogens [41], while in utero exposure to diethylstilbestrol is recognized to cause cancer in exposed offspring [42]. Likewise, studies within the field of fertility treatment (which includes the use of drugs such as progesterone) found an increased mortality risk in infants conceived by assisted reproduction techniques. A nationwide study assessed infant (<1 year) and childhood (1–18 years) mortality in singletons conceived through assisted reproductive techniques vs. naturally conceived singletons and reported an increased risk of infant mortality from birth to 1 year of life, predominantly in the early neonatal period [43]. Finally, we previously reported increased morbidities in children after in utero exposure to hormonal contraception [16,44,45,46].

Previously, we demonstrated that leukemia and ADHD were more common in children exposed to maternal hormonal contraception use [16,45], which may explain our findings of an excess mortality since both leukemia and ADHD are risk factors for early death in children [47,48]. Children with leukemia were additionally found to have a higher risk of death if exposed to maternal hormonal contraception use during pregnancy, and this could support the notion of childhood leukemia being a potential contributing cause to our observations. Furthermore, injuries are a leading cause of death overall in children [49], and severe injuries are associated with ADHD [50]. Thus, it is possible that overall child mortality is higher among children exposed to maternal use of hormonal contraception through an increased risk of cancer and ADHD-related injuries.

The finding that in utero exposure to hormonal contraception seemed to be a predictor for survival after childhood leukemia adds to our understanding of hormones being factors potentially not exclusively involved in the development of childhood leukemia [16] but, possibly, also in the survival after childhood leukemia. This could indicate that leukemias with a hormonal etiology have a different prognosis than leukemias that develop unrelated to in utero exposure to hormonal contraception. However, it could also be speculated that mothers who inadvertently become pregnant due to irregular hormonal contraception use may also be less compliant to the leukemia treatment offered to their children. However, irrespective of the underlying cause, this finding is important since it can be used to identify a group of children with poorer survival where closer surveillance and support can be needed. As an argument against the assumption of a non-biological explanation for our findings are the stronger associations found with a specific type of hormonal contraception used and also specifically in children with leukemia. If our finding was caused by bias, we would have expected a similar increase in mortality in children with CNS tumors despite a less demanding treatment regimen for parents to adhere to. This was not confirmed by our results; in contrast, the HRs were generally below unity.

The strengths of our study include the long follow-up time, accumulating to 16,099,288 person-years and 7390 number of deaths, which allowed for detailed assessments of children exposed in utero to different types of hormonal contraceptives. The validity and completeness of exposure and outcome are high, as they are based on information from nationwide population-based registries of high-quality and completeness. The sensitivity analyses showed that any unknown confounder would have to be unusually strong to explain our findings of increased mortality in exposed children, in particular, for in utero exposure to non-oral progestin-only products and in children with leukemia exposed in utero to hormonal contraception [34]. Within this field of research, a third variable that affects both exposure and outcome each by 2- or 3-fold is extremely uncommon [34], though we cannot exclude such a factor exists. Recently, a systematic review and meta-analysis [51] reported that lower parental education was a risk factor for child mortality, even after controlling for other markers of family socioeconomic status. In this study, we did account for parental education. Moreover, we assessed the effects of several potential confounders by change in estimate methods, but none of the potential confounders changed the risk estimate for death in the children. Moreover, parental education is found to be a good proxy for socioeconomic status in Denmark, and information on parental educational level in childhood has previously been used as a measure of childhood socioeconomic status [52]. As social inequality in multimorbidity has been reported, people of low socioeconomic status have a higher likelihood of having multimorbidity compared with individuals of higher socioeconomic status [52]. Hence, when adjusting for parental education level, it can be argued that we have (to some extent) indirectly adjusted for parental comorbidities. However, the results did not change after adjustment for parental education. Though residual or unknown cofounding may limit observational studies, observational studies are the only possible way to assess this research question in humans since randomized trials are not an option. To avoid the problem that never users of hormonal contraception represent a small subpopulation of women that could lead to confounding, we used children born of maternal “previous users” as the reference group. This group is likely more comparable to children of mothers with “recent” use or use “during” pregnancy. Thus, it is “the timing of exposure to hormonal contraception” in relation to pregnancy that is the main exposure of interest. A limitation of our study is that some women may have redeemed a prescription of hormonal contraception without taking the product. If so, this would cause an underestimation of the associations assessed, consequently leading to conservative estimates. Another is that the case-fatality analyses were limited by few exposed children with cancer. Thus, the case-fatality analyses should be assessed in larger populations of children with cancer to compute more robust estimates and to test these novel findings in another population. However, our main findings are expected to have good generalizability, and the results are expected to be broadly applicable, considering the nationwide and population-based study design.

## 5. Conclusions

In this study, in utero exposure to hormonal contraception was associated with increased mortality in Danish children and in Danish children diagnosed with leukemia. This was particularly pronounced for the non-oral progestin-only products.

## Figures and Tables

**Figure 1 cancers-15-03163-f001:**
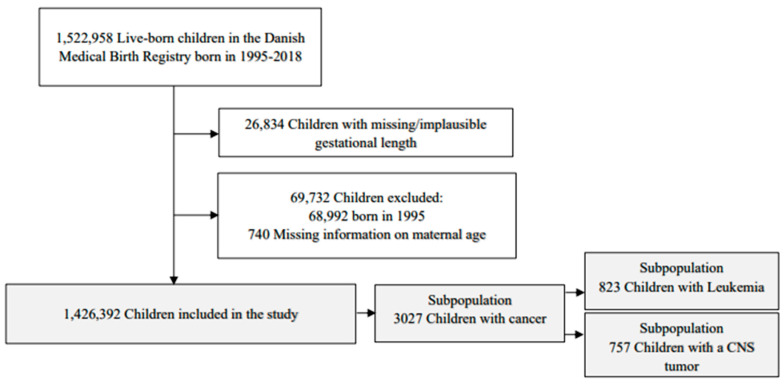
Identification of the study cohort and the sub-cohorts for case fatality analysis.

**Table 1 cancers-15-03163-t001:** Characteristics of the study population.

Characteristics	No. (%)			
	During Pregnancy	Recent Use	Previous Use	No Use
Number of children	17,513 (1.2) ^a^	154,381 (10.8) ^b^	952,800 (66.8) ^c^	301,698 (21.2)
Child characteristics				
Year of birth				
1996–1999	4081 (23.3)	26,532 (17.2)	140,434 (14.7)	152,919 (50.7)
2000–2004	4075 (23.3)	31,500 (20.4)	223,118 (23.4)	60,316 (20.0)
2005–2009	4070 (23.2)	35,194 (22.8)	244,180 (25.6)	33,190 (11.0)
2010–2014	3502 (20.0)	34,905 (22.6)	214,726 (22.5)	31,521 (10.5)
2015–2018	1785 (10.2)	26,250 (17.0)	130,342 (13.7)	23,752 (7.9)
Median (IQR)	2006 (2001–2012)	2008 (2003–2014)	2008 (2003–2013)	2000 (1997–2007)
Sex				
Male	9126 (52.1)	79,427 (51.5)	487,390 (51.2)	154,746 (51.3)
Female	8362 (47.8)	74,698 (48.4)	463,728 (48.7)	146,055 (48.4)
Missing	25 (0.1)	256 (0.2)	1673 (0.2)	897 (0.3)
Birth order				
First	7954 (45.4)	68,250 (44.2)	433,755 (45.5)	111,685 (37.0)
Second or higher	9553 (54.6)	86,100 (55.8)	518,861 (54.5)	189,739 (62.9)
Missing	6 (0.03)	31 (0.02)	184 (0.02)	274 (0.09)
Parental characteristics				
Origin (mother)				
Danish or descendant ^d^	15,500 (88.5)	142,316 (92.2)	883,169 (92.7)	187,473 (62.1)
Immigrant	2013 (11.5)	12,065 (7.8)	69,631 (7.3)	114,208 (37.9)
Missing	0 (0.0)	0 (0.0)	0 (0.0)	17 (0.01)
BMI ^$^ (mother)				
<25	6281 (35.9)	66,015 (42.8)	411,330 (43.2)	64,619 (21.4)
25–30	2563 (14.6)	26,333 (17.1)	157,974 (16.6)	23,694 (7.9)
31–35	1048 (6.0)	9038 (5.9)	58,293 (6.1)	8066 (2.7)
>35	571 (3.3)	4587 (3)	32,594 (3.4)	3990 (1.3)
Missing	7050 (40.3)	48,408 (31.4)	292,609 (30.7)	201,329 (66.7)
Maternal education ^e^				
Basic	6808 (38.9)	34,668 (22.5)	166,285 (17.5)	68,800 (22.8)
Vocational	6986 (39.9)	65,345 (42.3)	394,412 (41.4)	101,612 (33.7)
Higher	3136 (17.9)	51,767 (33.5)	380,941 (40.0)	81,629 (27.1)
Missing	583 (3.3)	2601 (1.7)	11,162 (1.2)	49,657 (16.5)
Paternal education ^e^				
Basic	5715 (32.6)	32,933 (21.3)	171,800 (18.0)	64,844 (21.5)
Vocational	8043 (45.9)	74,904 (48.5)	448,764 (47.1)	117,200 (38.9)
Higher	2793 (16.0)	41,604 (27.0)	299,286 (31.4)	80,600 (26.7)
Missing	962 (5.5)	4940 (3.2)	32,950 (3.5)	39,054 (12.9)
Maternal age at birth (y)				
<28	8781 (50.1)	60,522 (39.2)	282,079 (29.6)	84,850 (28.1)
28–31	4261 (24.3)	47,993 (31.1)	320,411 (33.6)	83,774 (27.8)
32–35	2841 (16.2)	31,160 (20.2)	229,328 (24.1)	76,828 (25.5)
>35	1630 (9.3)	14,706 (9.5)	120,982 (12.7)	56,246 (18.6)
Missing	0 (0.0)	0 (0.0)	0 (0.0)	0 (0.0)
Median (IQR)	27 (23–32)	29 (26–32)	30 (27–33)	31 (27–34)
Paternal age at birth (y)				
<28	5949 (34.0)	37,332 (24.2)	164,177 (17.2)	43,477 (14.4)
28–31	4109 (23.5)	44,665 (28.9)	273,430 (28.7)	67,561 (22.4)
32–35	3352 (19.1)	37,605 (24.4)	261,277 (27.4)	77,958 (25.8)
>35	3779 (21.6)	33,552 (21.7)	242,786 (25.5)	105,933 (35.1)
Missing	324 (1.9)	1227 (0.8)	10,990 (1.2)	6769 (2.2)
Median (IQR)	30 (26–35)	31 (28–35)	32 (29–36)	33 (30–38)
Maternal smoking ^f,^*	4577 (26.1)	25,027 (16.2)	137,865 (14.5)	40,195 (13.3)
Missing	1405 (8.0)	9361 (6.1)	49,933 (5.2)	48,395 (16.0)
Maternal infertility ^g^	505 (2.9)	7589 (4.9)	85,916 (9)	29,323 (9.7)
Missing	0 (0.0)	0 (0.0)	0 (0.0)	0 (0.0)

^a^ During pregnancy refers to use during pregnancy (first and second trimester). ^b^ Recent use refers to use 3 months or less before pregnancy start (except for non-oral progestin-only products (for further specification, see Section 2)). ^c^ Previous use refers to use more than 3 months before pregnancy start (except for non-oral progestin-only products (for further specification, see Section 2)). ^d^ Defined as having 2 parents without Danish citizenship and who were not born in Denmark. ^e^ Highest attained education before birth of the child. Basic indicates mandatory school grades 9–10; vocational, secondary school and vocational education; higher, short-, medium-, and long-term higher education. ^f^ Maternal smoking measured in the first trimester of pregnancy. * Information on maternal smoking was only available from 1998 onwards. ^g^ ICD-8 code 628* and ICD-10 code N97*. ^$^ Information on BMI was only available from 2004 onwards. Abbreviations: BMI = body mass index; wk = weeks; No = number; IQR = interquartile range; ICD = International Classification of Diseases (ICD) codes version ICD-10.

**Table 2 cancers-15-03163-t002:** Long-term mortality according to maternal use of any type of hormonal contraception.

	Number of Children	Person-Years of Follow-Up	Long-Term Mortality			
			Number of Deaths	Mortality Rate per 100,000 (95% CI) ^$^	HR (95% CI) *	*p*-Value
Previous use	952,800	9,908,070	4420	66.3 (63.5–69.1)	1 (reference)	
Recent use	154,381	1,578,630	748	67.1 (62.5–73.2)	1.02 (0.94–1.11)	0.613
During pregnancy	17,513	204,490	114	80.1 (65.4–96.6)	1.22 (1.01–1.48)	0.040

Maximum age at the longest follow-up was 23 years. Recent use refers to use 3 months or less before pregnancy start (except for non-oral progestin-only products (for further specification, see Section 2)). Previous use refers to use more than 3 months before pregnancy start (except for non-oral progestin-only products (for further specification, see Section 2)). ^$^ Adjusted for year of birth (categories: 1996–1999, 2000–2004, 2005–2009, 2010–2014, 2015–2018) and maternal age at birth (categories: <28, 28–31, 32–35, >35). * Stratified by year of birth (above mentioned categories) and maternal age at birth (above mentioned categories). HR = Hazard ratio.

**Table 3 cancers-15-03163-t003:** Long-term mortality according to maternal use of specific types of hormonal contraception.

	Number of Children	Person-Years of Follow-Up	Long-Term Mortality			
			Number of Deaths	Mortality Rate per 100,000 (95% CI) ^$^	HR (95% CI) *	*p*-Value
Any type						
Previous use	952,800	9,908,070	4420	66.3 (63.5–69.1)	1 (reference)	…
Combined contraceptives						
Oral contraception						
Recent use	111,608	1,314,780	585	68.2 (62.1–74.4)	1.04 (0.95–1.13)	0.424
During pregnancy	14,746	176,610	97	80.4 (63.5–97.3)	1.22 (0.99–1.50)	0.064
Non-oral contraceptives						
Recent use	3138	23,210	12	61.3 (23.7–98.8)	0.93 (0.51–1.71)	0.815
During pregnancy	501	3720	<5 ^£^	30.8 (-29.6–912)	0.47 (0.66–3.32)	0.447
Progestin-only contraceptives						
Oral contraception						
Recent use	7636	63,770	27	55.4 (34.4–76.4)	0.82 (0.56–1.19)	0.296
During pregnancy	1405	12,950	7	72.6 (18.6–126.5)	1.08 (0.51–2.26)	0.845
Non-oral contraceptives						
Recent use	33,547	176,880	132	72.4 (60.0–84.8)	1.04 (0.87–1.24)	0.671
During pregnancy	377	2520	5	203.9 (23.9–384.0)	2.92 (1.21–7.04)	0.017
Emergency contraceptives						
Recent use	1203	20,650	6	50.5 (10.0–91.1)	0.77 (0.34–1.71)	0.515
During pregnancy	591	10,240	5	87.2 (10.5–163.8)	1.31 (0.55–3.15)	0.544

Maximum age at the longest follow-up was 23 years. During pregnancy refers to use during pregnancy. Recent use refers to use 3 months or less before pregnancy start (except for non-oral progestin-only products (for further specification, see Section 2)). Previous use refers to use more than 3 months before pregnancy start (except for non-oral progestin-only products (for further specification, see Section 2)). ^$^ Adjusted for year of birth (categories: 1996–1999, 2000–2004, 2005–2009, 2010–2014, 2015–2018) and maternal age at birth (categories: <28, 28–31, 32–35, >35). * Stratified by year of birth (above mentioned categories) and maternal age at birth (above mentioned categories). ^£^ Exact number <5 or equivalent is blinded according to the interpretation of the General Data Protection Regulation by Statistics Denmark. HR = Hazard ratio.

**Table 4 cancers-15-03163-t004:** Long-term mortality in children who developed any type of cancer, leukemia or CNS tumor according to maternal use of any type of hormonal contraception.

	Number of Children	Person-Years of Follow-Up	Long-Term Mortality			
Any cancer			Number of Deaths	Mortality Rate per 1000 (95% CI) ^$^	HR (95% CI) *^,¤^	*p*-Value
Previous use	1864	260	217	19.7 (16.7–22.7)	1 (reference)	…
Recent use	304	11,590	42	24.6 (16.6–32.7)	1.18 (0.85–1.64)	0.325
During pregnancy	43	1840	7	30.7 (6.2–552.9)	1.49 (0.70–3.17)	0.299
Leukemia						
Previous use	548	80	51	16.2 (9.7–22.6)	1 (reference)	…
Recent use	86	3980	12	26.0 (8.8–43.2)	1.35 (0.69–2.63)	0.379
During pregnancy	13	620	<5 ^£^	65.1 (−15.1–145.3)	3.62 (1.33–9.87)	0.012
CNS tumors						
Previous use	446	70	82	37.0 (27.5–46.5)	1 (reference)	…
Recent use	81	2350	12	31.0 (12.3–49.6)	0.83 (0.45–1.54)	0.555
During pregnancy	9	410	<5 ^£^	17.8 (−18.9–54.4)	0.65 (0.08–5.28)	0.688

Maximum age at the longest follow-up was 23 years. During pregnancy refers to use during pregnancy. Recent use refers to use 3 months or less before pregnancy start (except for non-oral progestin-only products (for further specification, see Section 2)). Previous use refers to use more than 3 months before pregnancy start (except for non-oral progestin-only products (for further specification, see Section 2)). ^$^ Adjusted for year of birth (categories: 1996–1999, 2000–2004, 2005–2009, 2010–2014, 2015–2018) and maternal age at birth (categories: <28, 28–31, 32–35, >35). * Stratified by year of birth (above mentioned categories) and age of the children at cancer diagnosis (<1 year old, 1–5 years old, 6–10 years old, and >10 years old). ^¤^ Analyses were further adjusted for maternal age at birth (above mentioned categories). ^£^ Exact number <5 or equivalent is blinded according to the interpretation of the General Data Protection Regulation by Statistics Denmark. HR = Hazard ratio.

## Data Availability

Data are stored remotely on a secure platform at Statistics Denmark. According to Danish regulations, individual-level data can only be made available for researchers who fulfill legal requirements for access to sensitive data. Please contact Lina Mørch (morch@cancer.dk) for further questions about data access.

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
