# Peer review of "In Utero Exposure to Hormonal Contraception and Mortality in Offspring with and without Cancer: A Nationwide Cohort Study"

_cancers, 2023, doi:10.3390/cancers15123163_

Round 1

Reviewer 1 Report

Line 180-; Children born to mothers with previous exposure to contraceptive hormones account for 67% of the cases, but would the results be different if these 952800 cases were excluded and compared to children born to pregnant women with no prior exposure to hormones?

How much fetal toxicity has been reported for each hormone agent?

Are the fetal toxicities or birth outcomes from the hormones detected in this study consistent with or divergent from previously reported adverse events for the hormones in question?

The statistical analysis is fine, and I have no objection to the study report, but I think a more careful determination of causality is needed (since I do not think all the confounding factors for the birth outcomes reported here have been taken into account).

Line 340-"In utero exposure to hormonal contraception was associated with increased child mortality"

It might be better to say that it may be associated with death from disease.

Also, although statistically significant, the observed frequencies are small, so caution should be exercised in judging this result.

Author Response

Please see the attatchment. 

Reviewer 2 Report

The manuscript (ID: cancers-2425457) aimed to assess overall child mortality as well as prognosis in children with cancer after in utero exposure to hormonal contraception.

In this paper, it is necessary to enter several corrections (major revision).  

Some of the comments are:      

  • Lines 36-38: Provide data on the frequency of hormonal contraceptive use in women in Denmark. 
  • Lines 43-45: In addition to the accident with the use of thalidomide (sedative) during pregnancy, the results of previous studies that indicated the connection between the use of hormones during pregnancy and cancers should be mentioned. There are studies that state the connection of diethylstilbestrol and cancers. Diethylstilbestrol (DES), a synthetic form of the female hormone estrogen, that was prescribed to pregnant women between 1940 and 1971 to prevent miscarriage, premature labor, and related complications of pregnancy. The use of DES declined after studies linked prenatal (in utero) DES exposure to a type of cancer of the cervix and vagina, as well as to other effects. Females exposed to DES in utero, commonly called DES daughters, are at increased risk of several specific cancers. The references are as follows:
    • Al Jishi T, Sergi C. Current perspective of diethylstilbestrol (DES) exposure in mothers and offspring. Reproductive Toxicology 2017; 71:71–77.
    • Herbst AL, Ulfelder H, Poskanzer DC. Adenocarcinoma of the vagina. Association of maternal stilbestrol therapy with tumor appearance in young women. The New England Journal of Medicine 1971; 284(15):878–881.
    • Giusti RM, Iwamoto K, Hatch EE. Diethylstilbestrol revisited: A review of the long-term health effects. Annals of Internal Medicine 1995; 122(10):778–788.
    • Palmer JR, Wise LA, Hatch EE, et al. Prenatal diethylstilbestrol exposure and risk of breast cancer. Cancer Epidemiology, Biomarkers & Prevention 2006; 15(8):1509–1514.
    • Troisi R, Hatch EE, Titus L, et al. Prenatal diethylstilbestrol exposure and cancer risk in women. Environmental and Molecular Mutagenesis 2019; 60(5):395–403.
    • Verloop J, van Leeuwen FE, Helmerhorst TJ, van Boven HH, Rookus MA. Cancer risk in DES daughters. Cancer Causes and Control 2010; 21(7):999–1007.
    • Huo D, Anderson D, Palmer JR, Herbst AL. Incidence rates and risks of diethylstilbestrol-related clear-cell adenocarcinoma of the vagina and cervix: Update after 40-year follow-up. Gynecologic Oncology 2017; 146(3):566–571.
    • Hoover RN, Hyer M, Pfeiffer RM, et al. Adverse health outcomes in women exposed in utero to diethylstilbestrol. New England Journal of Medicine 2011; 365(14):1304–1314.
    • Troisi R, Hyer M, Titus L, et al. Prenatal diethylstilbestrol exposure and risk of diabetes, gallbladder disease, and pancreatic disorders and malignancies. Journal of Developmental Origins of Health and Disease 2021; 12(4):619–626.
    • Strohsnitter WC, Hyer M, Bertrand KA, et al. Prenatal diethylstilbestrol exposure and cancer risk in males. Cancer Epidemiology, Biomarkers & Prevention 2021; 30(10):1826–1833.
    • Strohsnitter WC, Noller KL, Hoover RN, et al. Cancer risk in men exposed in utero to diethylstilbestrol. Journal of the National Cancer Institute 2001; 93(7):545–551.
    • Kaufman RH, Adam E. Findings in female offspring of women exposed in utero to diethylstilbestrol. Obstetrics and Gynecology 2002; 99(2):197–200.
  • Lines 48-49: Provide data on incidence and mortality from cancer in children in Denmark: for example, data for the most frequent localizations of malignant tumors, with a description of trends, with comparisons with incidence/mortality and trends in other countries (citing appropriate references).
  • Lines 49-51: Provide the appropriate reference for the claim in this sentence.
  • Line 55: Add a new subsection `Study design', with an appropriate description of the study design applied in this research. 
  • Lines 56-117: Make this text of the Methodology section more clear and transparent by describing in special subsections `Study population', `Data sources', `Measures' and `Variables' in this manuscript. In this section about methodology of this manuscript, provide data on the quality of all the data used in this study.
  • Figure 1 and Figure S1: Combine these 2 Figures into one Figure, for the reason that half of the data is repeated on Figure S1 (unnecessary), as well as for better data transparency.  
  • Lines 269-275: It is a pleasure to see that in the first chapter of the Discussion section, the authors highlighted the most important results presented in this paper.
  • Lines 307-330: Move the data quality information to the methodology section of this manuscript. In the text about the strength of the study, it is enough to mention the quality of the data, without details. Anyone responsible for the quality of the data should be listed in the study methodology section. In the discussion about the strength of the study, it should be highlighted primarily what the authors of this paper did themselves.  
  • Lines 331-339: Limitations of the study should be improved. First, the majority of HR estimates were determined for children among mothers who used hormonal contraception (as the reference category), but only a small number of HR estimates were conducted according to the reference category of mothers who did not use hormonal contraception. Therefore, it is necessary to discuss the circumstances when all comparisons of HR were carried out according to the reference category made by mothers without using hormonal contraception, would this lead to an underestimation or overestimation of HR? Furthermore, discuss the possible influence on the assessment of HR and some other factors/variables that were not included in this paper, such as the personal medical history of the mother and father (such as comorbidity of the mother and father in connection with hormonal disorders, infections, cancers, etc.), family history of cancer, socio-economic status, other exposures, etc. Also, discuss the duration of the study in relation to the age of all children included in the study: are children born at the end of the study period and children born at the beginning of the study period at the same risk in terms of duration of exposure and in terms of outcomes? Finally, the duration of taking hormonal contraception is descriptively categorized in this manuscript, without precise determination of the real duration (in months/years, with or without interruptions in use, etc.). Discuss this.     

The quality of English language is appropriate.  

Author Response

Please see the attatchment.

Round 2

Reviewer 2 Report

Thank you for the opportunity to re-review manuscript ID: cancers-2425457. Overall, the authors submitted a version of the manuscript for re-review in which significant corrections were made. The authors satisfactorily answered all my comments and provided appropriate explanations. All subsections of the paper have been significantly improved, from the sections Introduction, Methods, Discussion, to the list of References, which have been supplemented with appropriate references.

Thanks to the authors for the significant effort they put into the preparation of the work. A good job was done. 

My recommendation is: Accept.  

The quality of English language is appropriate.